# Bevacizumab-Induced Thrombotic Microangiopathy (TMA) in Metastatic Lung Adenocarcinoma Patients Receiving Nivolumab Combined with Bevacizumab, Carboplatin and Paclitaxel: Two Case Reports

**Ping-Chih Hsu [1,2] , Tai-Di Chen [2,3] , Tsung-Yu Tsai [3,4] and Cheng-Ta Yang [1,5,6,*]**

1. Department of Thoracic Medicine, Chang Gung Memorial Hospital at Linkou, Taoyuan City 33305, Taiwan
2. Department of Medicine, College of Medicine, Chang Gung University, Taoyuan City 33302, Taiwan
3. Department of Pathology, Chang Gung Memorial Hospital at Linkou, Taoyuan City 33305, Taiwan
4. Department of Nephrology, Kidney Research Center, Chang Gung Memorial Hospital at Linkou, Taoyuan City 33305, Taiwan
5. Department of Internal Medicine, Taoyuan Chang Gung Memorial Hospital, Taoyuan City 33378, Taiwan
6. Department of Respiratory Therapy, College of Medicine, Chang Gung University, Taoyuan City 33302, Taiwan
* Correspondence: yang1946@cgmh.org.tw; Tel.: +886-3-3281200 (ext. 8468); Fax: +886-3-3278211

**Abstract:** Anti-programmed death-1 (PD-1)/programmed death ligand 1 (PD-L1) immune checkpoint inhibitors (ICIs), combined with bevacizumab and platinum-based chemotherapy, have shown promising efficacy in treating metastatic non-squamous cell lung cancer in phase 3 clinical trials. However, drug-induced nephrotoxicity is an uncommon but threatening adverse effect when using this combination therapy, and should be evaluated and managed carefully. Here, we present two patients experiencing late-onset asymptomatic heavy proteinuria during the clinical trial. Kidney biopsies performed finally identified bevacizumab-induced thrombotic microangiopathy (TMA), and the proteinuria was decreased after discontinuing bevacizumab permanently. Our report suggests that a kidney biopsy is needed for those receiving ICIs in combination with bevacizumab and chemotherapy and experiencing nephrotoxicity such as heavy proteinuria.

**Keywords:** bevacizumab; proteinuria; thrombotic microangiopathy (TMA); immunotherapy; nivolumab; ant-PD-1; non-small cell lung cancer

## 1. Clinical Practice Points

- Drug-induced nephrotoxicity is an uncommon but threatening adverse effect when using ICIs combined with bevacizumab and chemotherapy.
- A kidney biopsy is indicated for metastatic non-squamous cell lung cancer patients experiencing nephrotoxicity during this combination therapy.

## 2. Introduction

The therapy of nivolumab in combination with carboplatin, paclitaxel, and bevacizumab was shown to be promising in the first-line treatment of advanced non-squamous, non-small-cell lung cancer (NSCLC) in a previous pivotal clinical trial ONO-4538-52/TASUKI-52 (61.5% objective response rate, 12.1 months of median progression-free survival) [1]. We should be aware of some adverse events (AEs) from this combination therapy, and an appropriate determination of the drug that induced the AE is important for its management [2]. Here, we investigate bevacizumab-induced thrombotic microangiopathy (TMA) in two metastatic NSCLC patients receiving nivolumab combined with bevacizumab, carboplatin, and paclitaxel, and both patients experienced a survival benefit from the combination treatment.

## 3. Case Presentation

The clinical treatment and follow-up courses of the two patients are summarized in Figure 1. The first patient was a 59-year-old male who had a major medical history of an acute coronary artery event, that was treated with angioplasty with a stent implantation five years before the lung cancer. He was receiving aspirin therapy only after the acute coronary artery event and did not have recorded hypertension or symptoms of heart failure. He was diagnosed with stage IVa right upper lung adenocarcinoma with solitary cerebellar metastasis in July 2017. The neurological symptoms were stabilized after brain tumor surgery and radiotherapy. He was enrolled in a phase 3 trial, and combination therapy regimens of nivolumab combined with bevacizumab, carboplatin, and paclitaxel were administrated from September 2018 to January 2019. A partial response (PR) was achieved from the four combination treatment regimens, and the therapy was switched to bevacizumab combined with nivolumab every three weeks as maintenance therapy. This patient received regular dipstick urine analysis (UA), a hemogram, and biochemistry tests during every cycle of therapy. Proteinuria was detected by dipstick UA in the 41st cycle of therapy (dipstick proteinuria 3+). Proteinuria (dipstick UA 3+) with normal serum creatinine (0.65 mg/dL) was detected in the 41st cycle of therapy in February 2021. There were no recorded clinical symptoms such as fever, skin rashes, gross hematuria, decreased urine amount, or edema associated with the appearance of proteinuria. Twenty-four-hour total urine protein showed heavy proteinuria of 3.52 g/day. A kidney ultrasonography performed did not show significant renal parenchymal abnormality (Supplementary Figure S1A). A kidney biopsy was performed by the referral nephrologist to determine the cause of proteinuria, and the diagnosis of TMA was finally confirmed (Figure 2A–C). Bevacizumab was discontinued permanently because of possible bevacizumab-induced TMA, and single nivolumab therapy was continued. The dipstick UA showed negative proteinuria after discontinuing bevacizumab.

A 74-year-old male patient without past major medical and medication history was diagnosed with stage IVa left upper lung adenocarcinoma with bilateral lung metastasis in May 2019. He began combination therapy regimens of nivolumab combined with bevacizumab, carboplatin, and paclitaxel in a phase 3 clinical trial in June 2019. He had a partial response (PR) to six cycles of nivolumab combined with bevacizumab, carboplatin, and paclitaxel, and this was followed by bevacizumab combined with nivolumab every three weeks as maintenance therapy. Regular dipstick urine analysis (UA), a hemogram, and biochemistry tests were performed during every cycle of therapy to follow the trial protocol. Proteinuria (dipstick UA 3+) with normal serum creatinine (1.08 mg/dL) was detected at the 20th cycle of therapy in August 2020. The patient had no recorded clinical symptoms such as fever, skin rashes, gross hematuria, decreased urine amount, or edema associated with the proteinuria. Twenty-four-hour total urine protein was 3.96 g/day, and the kidney ultrasonography did not show significant renal parenchymal abnormality (Supplementary Figure S1B). A kidney biopsy performed by the referral nephrologist to determine the cause of proteinuria confirmed the diagnosis of TMA (Figure 2D–F). Bevacizumab was discontinued permanently, and the dipstick UA showed negative proteinuria in December 2020. Single nivolumab therapy was continued after the discontinuation of bevacizumab. The changes in proteinuria, serum creatinine level, and treatment agents in both patients are shown in Figure 3. Both patients were recorded with a grade I hypertension AE during bevacizumab combined with nivolumab maintenance therapy, but no anti-hypertensive agent was administrated. There was no other immune-mediate AE recorded in these two patients.

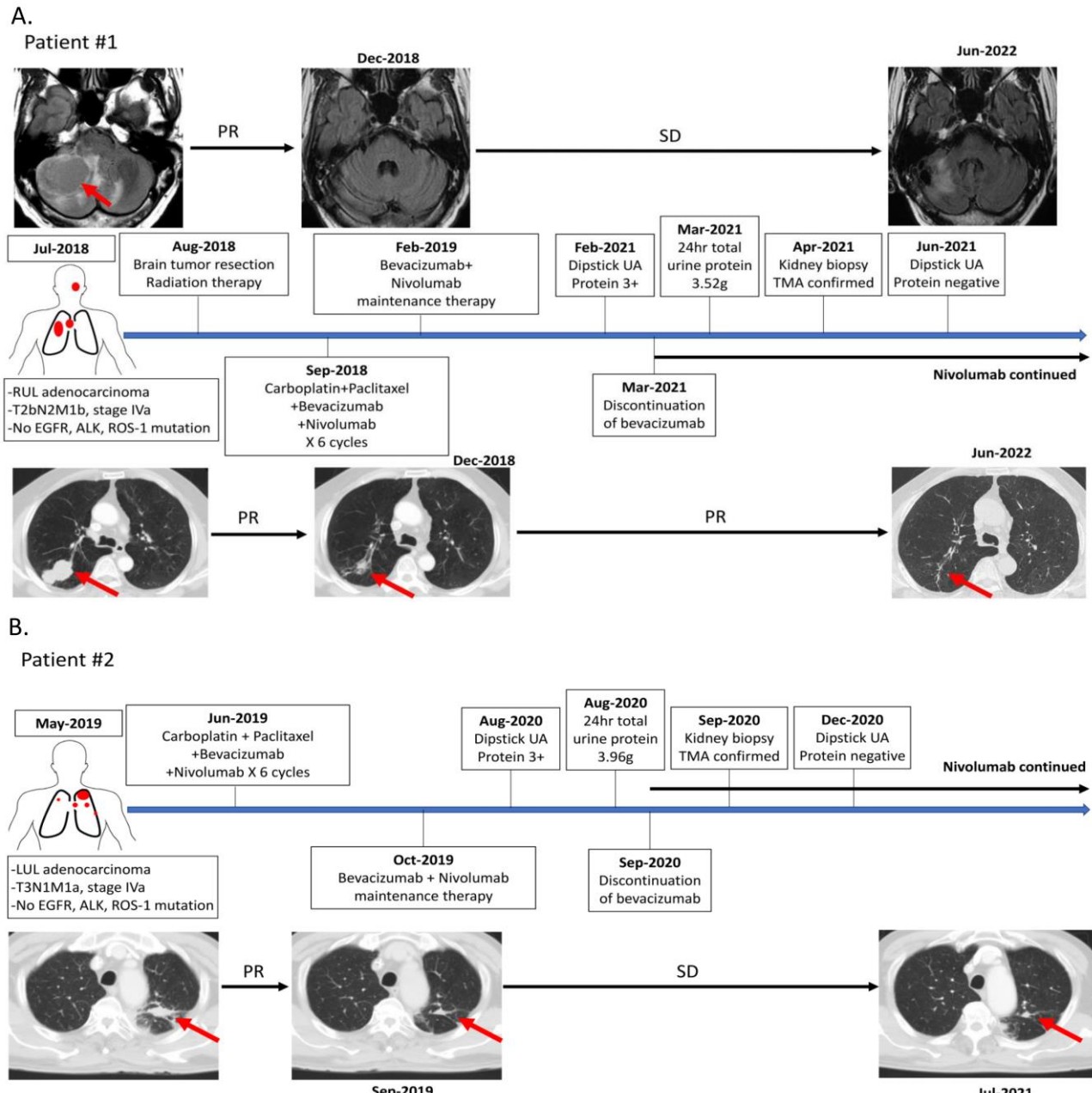

**Figure 1.** The treatment and follow-up timeline summary of 2 patients. The primary lung cancer and metastatic tumor are labeled by red arrow. (**A**) The treatment and follow-up timeline summary of patient #1. (**B**) The treatment and follow-up timeline summary of patient #2. Abbreviations: EGFR, epidermal growth factor receptor; ALK, anaplastic lymphoma kinase; PR, partial response; SD, stable disease; UA, urine analysis.

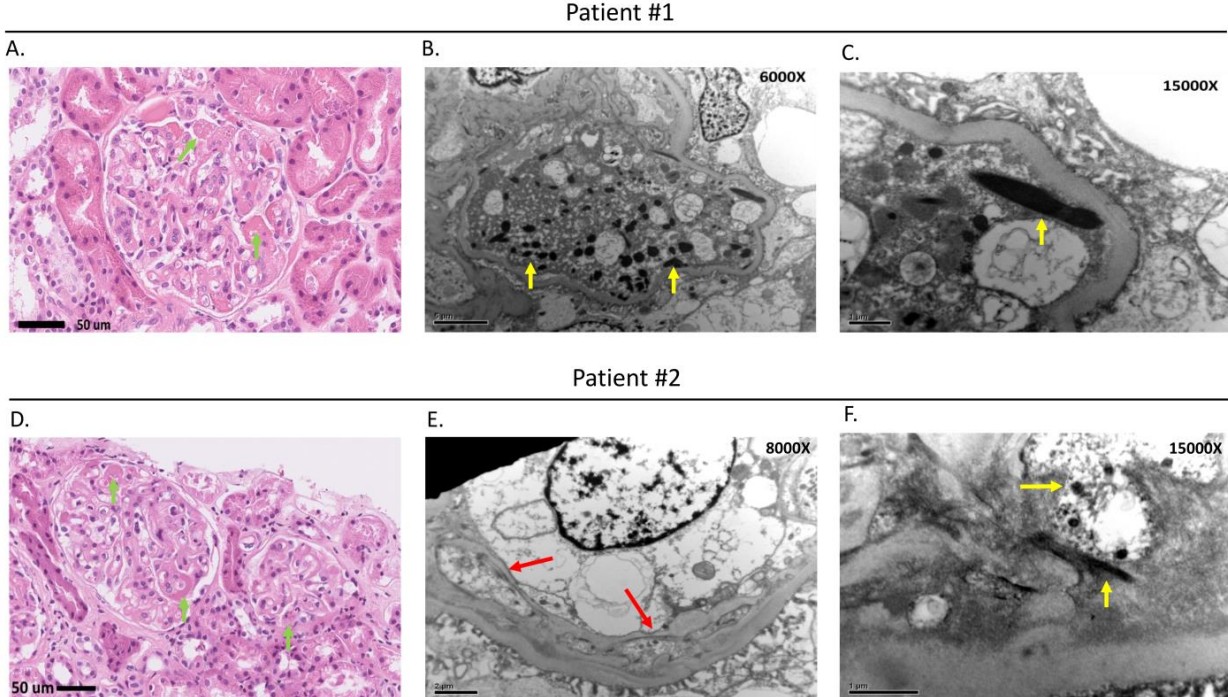

**Figure 2.** Kidney biopsy pathological findings for the 2 patients. (**A**,**D**) Light microscopy showing fibrin thrombi and tactoids (green arrows) in the glomerular capillary. (**B**,**C**,**F**) Electron microscopy showing glomerular injury with fibrin thrombi and tactoids (yellow arrows), and (**E**) double-contoured appearance of the newly formed basement membrane (red arrows). The biopsy pathological findings confirmed the diagnosis of thrombotic microangiopathy (TMA).

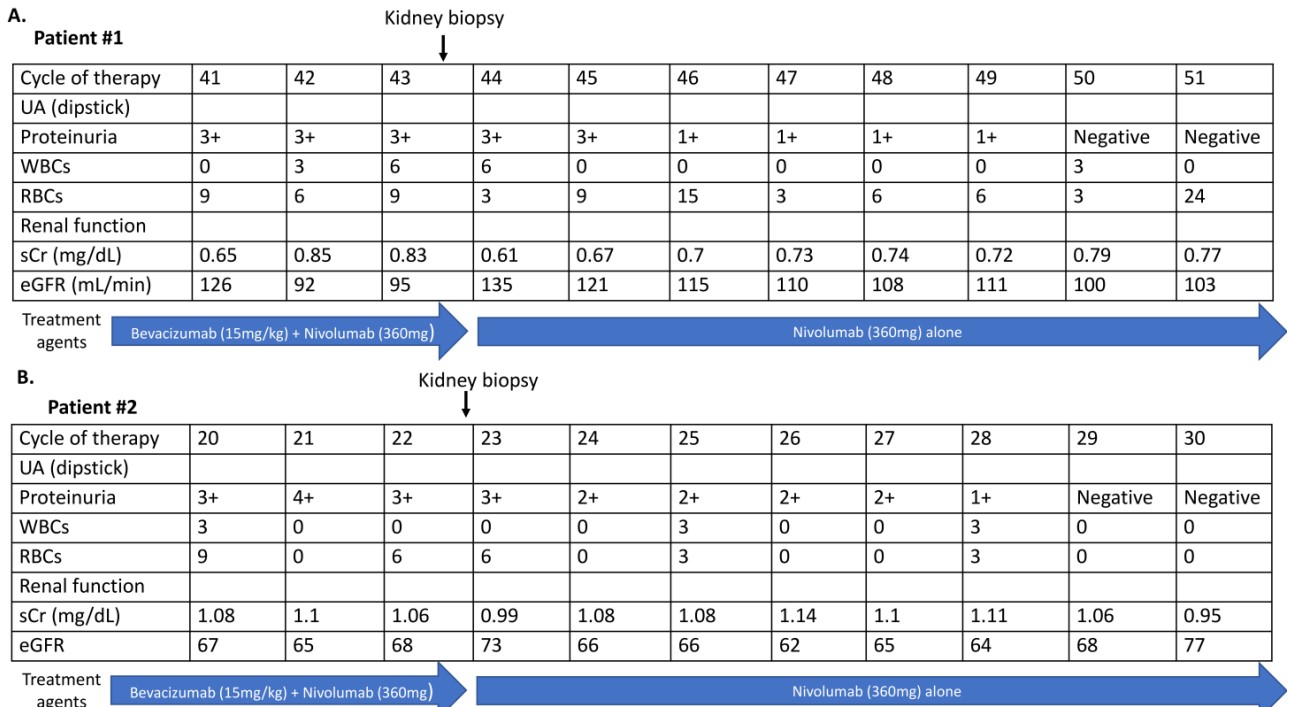

**Figure 3.** The changes in proteinuria detected by dipstick UA, renal function, and treatment agents in the 2 patients. (**A**) Patient #1, (**B**) Patient #2. Abbreviations: UA, urine analysis; WBCs, white blood cells; RBCs, red blood cells; sCr, serum creatinine; eGFR, estimated glomerular filtration rate.

## 4. Discussion

We were the first to report on bevacizumab-induced TMA with pathology in immune checkpoint inhibitor (ICI), combined with bevacizumab and chemotherapy in advanced non-squamous NSCLC. In a previous trial IMPOWER150, which explored the efficacy of atezolizumab combined with carboplatin, paclitaxel, and bevacizumab therapy in NSCLC, the adverse event of proteinuria occurred in 12.5% of the study subjects. The evaluation and management of serious proteinuria in the same trial were rarely reported [3]. A previous study found that the production of vascular endothelial growth factor (VEGF) by podocytes is required to maintain the adjacent glomerular endothelium in the in vivo experiments [4]. The disruption of VEGF function, either through genetic or pharmacologic ablation resulting in proteinuria and renal damage with a pattern of thrombotic microangiopathy, was shown in a mouse model. In the immunohistochemical staining analysis of a kidney biopsy from the mouse model, the complement components and immune complexes were negative, but fibrin staining was positive. The findings of this previous study suggested that the reduction of glomerular VEGF by bevacizumab therapy may be the pathophysiology of TMA [4]. In addition, this study demonstrated the classic pathological features of TMA in six patients who had proteinuria after bevacizumab therapy [4]. Another previous study also reported that TMA was the main biopsy finding in bevacizumab-induced proteinuria [5].

Though ICIs had been reported to induce TMA in previous studies, the incidence was low. In addition, the patients who were diagnosed with ICI-induced TMA presented with the clinical feature of acute kidney injury (AKI) with obviously increased serum creatinine and proteinuria that was not heavy [6,7]. The two patients in this report presented with asymptomatic heavy proteinuria, and did not exhibit typical symptoms of AKI such as increased serum creatinine level, oliguria, or anuria. Based on the previous study and clinical presentation of the two patients, bevacizumab was discontinued prior to nivolumab. The proteinuria decreased after discontinuation of bevacizumab. Although chemotherapy including carboplatin and paclitaxel may have nephrotoxicity, the two agents were administrated in the initial 6 cycles of therapy, and the proteinuria developed after cycle 20 of therapy in the two patients. The pathological findings for both patients showed fibrin thrombi, tactoids, and a double-contoured appearance of the capillary wall within the glomerular capillary and confirmed the diagnosis of TMA. Taken together, bevacizumab-induced TMA was confirmed for the two patients in this report.

Previous studies have reported that acute interstitial nephritis (AIN) is the most frequent pathological finding in ICI-induced nephrotoxicity [6–8]. In an analysis of a previous clinical study, immune-related adverse events and hypertension were independently predictive factors associated with ICI-induced AKI [7]. Bevacizumab in combination with chemotherapy or epidermal growth factor receptor–tyrosine kinase inhibitor (EGFR-TKI) for the treatment of advanced NSCLC has been explored in several prospective clinical trials, and the incidence rate of a grade 3–4 AE of proteinuria recorded in these previous trials ranged from 6% to 8% in patients who received bevacizumab therapy [9]. In two recent phase 3 trials investigating the efficacy of ICIs combined with bevacizumab, the incidence rate of a grade 3–4 AE of proteinuria ranged from 2.8% to 4.8% in the study arm with ICIs combined with bevacizumab, which was not higher than that in the control arm without ICIs (2.8–3.6%) [1,3]. The combination of ICIs and bevacizumab did not increase the risk of nephrotoxicity and proteinuria.

Our report suggests that regular UA and renal function tests are necessary in advanced NSCLC patients receiving ICIs combined with bevacizumab, and a kidney biopsy helps to determine the cause of drug-induced nephrotoxicity and proteinuria.

**Supplementary Materials:** The following supporting information can be downloaded at: https://www.mdpi.com/article/10.3390/clinpract13010018/s1, Figure S1: Kidney ultrasonography of 2 study patients. (A) Patient #1 (B) Patient #2.

**Author Contributions:** Acquisition of data, P.-C.H., T.-D.C. and T.-Y.T.; Writing, review, and revision of the manuscript, P.-C.H., T.-D.C. and C.-T.Y.; Study supervision, C.-T.Y. All authors have read and agreed to the published version of the manuscript.

**Funding:** This work was partially supported by the Chang-Gung Medical Research Project (CMRP No. CMRPG3G1661) and Taiwan Ministry of Science and Technology (MOST) (grant nos. 111-2628-B-182A-003 and 110-2628-B-182A-019 to P.C.H.).

**Institutional Review Board Statement:** The retrieval and analysis of information in the report were approved by Institutional Review Board (IRB) (202201732B0) of the Chang Gung Medical Foundation. Written informed consent was not required by IRB because of the retrospective nature. No identifiable information such as personal ID or birth date was reported in this manuscript.

**Informed Consent Statement:** Not required for retrospective study.

**Data Availability Statement:** Not applicable.

**Conflicts of Interest:** All authors have no conflicts of interest.

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
