# Peer review of "Bevacizumab-Induced Thrombotic Microangiopathy (TMA) in Metastatic Lung Adenocarcinoma Patients Receiving Nivolumab Combined with Bevacizumab, Carboplatin and Paclitaxel: Two Case Reports"

_clinpract, doi:10.3390/clinpract13010018_

Round 1

Reviewer 1 Report

Thank you for the opportunity to review the case report titled “ Bevacizumab-induced thrombotic microangiopathy (TMA) in metastatic lung adenocarcinoma patients receiving nivolumab combined with bevacizumab, carboplatin and paclitaxel: Two Case Reports”.  

After reviewing the cases, the most likely causes of proteinuria would be either bevacizumab related microangiopathy or interstitial nephritis from nivolumab.  By probability, the former is more likely than the latter.  But one cannot rule out the possibility of other concomitant medications.  In these 2 cases, no information has been provided on the co-morbidity and concurrent medications which may predispose the patients to microangiopathy or increase the complexity to the diagnosis.

In addition, there should be more details on the investigations done prior to the renal biopsy.  Typical interstitial nephritis does not present with heavy proteinuria, and there may be WBC casts in urine, eosinophilia, fever, rash etc while bevacizumab-related microangiopathy, unlike microangiopathy like TTP and HUS, does not have typical features of hypertension, CNS symptoms, hemolysis etc.  Any urinalysis done?  Any other laboratory abnormalities or clinical symptoms?  

Based on the modest novelty and the lack thereof details, it is hard to see the value of publication of these cases reports.

Author Response

Response to Reviewer 1 Comments

After reviewing the cases, the most likely causes of proteinuria would be either bevacizumab related microangiopathy or interstitial nephritis from nivolumab.  By probability, the former is more likely than the latter. But one cannot rule out the possibility of other concomitant medications.

Point 1: In these 2 cases, no information has been provided on the co-morbidity and concurrent medications which may predispose the patients to microangiopathy or increase the complexity to the diagnosis.

Response 1: Regarding the concern about “no information has been provided on the co-morbidity and concurrent medications, we added the past medical history of the two patients in revised manuscript as suggested.

The first patient is a 59-year-old male who had a major medical history of acute coronary artery event which was treated by angioplasty with stent implantation five years before lung cancer. He is receiving aspirin therapy only after acute coronary artery event, and did not have recorded hypertension or symptom of heart failure.

The second patient is a 74-year-old male patient who have no past major medical history.

Point 2: In addition, there should be more details on the investigations done prior to the renal biopsy. Typical interstitial nephritis does not present with heavy proteinuria, and there may be WBC casts in urine, eosinophilia, fever, rash etc while bevacizumab-related microangiopathy, unlike microangiopathy like TTP and HUS, does not have typical features of hypertension, CNS symptoms, hemolysis etc. Any urinalysis done?  Any other laboratory abnormalities or clinical symptoms? 

Response 2: In response to the concern about “there should be more details on the investigations done prior to the renal biopsy”, we added the data of urine analysis, renal function and kidney sonography in revised manuscript as suggested.

Patient#1 was not recorded clinical symptoms such as fever, skin rashes, gross hematuria, decreased urine amount or edema associated with the appearance of proteinuria. Twenty-four hours urine total protein examined showed heavy proteinuria by 3.52gram/day. A kidney ultrasonography performed did not show significant renal parenchymal abnormality (Supplementary figure S1A).

    Patient #2 had no recorded clinical symptoms such as fever, skin rashes, gross hematuria, decreased urine amount or edema associated with the proteinuria. Twenty-four urine total protein examined was 3.96gram/day, and kidney ultrasonography performed did not show significant renal parenchymal abnormality (Supplementary figure S1B).

The changes of proteinuria, serum creatinine level and treatment agents in both patients are shown in Figure 3 in revised manuscript.

Point 3: Based on the modest novelty and the lack thereof details, it is hard to see the value of publication of these cases reports.

Response 3: In response to the comment about “based on the modest novelty and the lack thereof details, it is hard to see the value of publication of these cases reports.”, we had added more detail clinical information of the 2 patients and extend the discussion section to address all reviewer’s comments.

The combination of bevacizumab and ICIs has been shown to improved survival of advanced lung cancer patients in recent clinical trials, and the use of this combination in clinical practice increases.

Our report suggests that regular UA and renal function tests are necessary in advanced NSCLC patients receiving ICIs combined with bevacizumab, and kidney biopsy is helpful to determine the cause of drug-induced nephrotoxicity and proteinuria.

Reviewer 2 Report

Some English spelling has to be corrected. You do not explain the possible  pathophysiology of this pharmacological complication, which could be referred to as secondary endothelial damage limited to glomeruli, or an immune reaction to a drug.  Correlated references could be added. Could kidney imaging, eventually US, be useful for an early diagnosis? The other renal functional tests are not reported. 

Author Response

Response to Reviewer 2 Comments

Point 1: Some English spelling has to be corrected.

Response 1: The English edition of revised manuscript had been done as suggested.

Point 2: You do not explain the possible pathophysiology of this pharmacological complication, which could be referred to as secondary endothelial damage limited to glomeruli, or an immune reaction to a drug. Correlated references could be added.

Response 2: In response to this comment, we added paragraph to discuss this point deeper and cited further references in revised manuscript as suggested.

A previous study found that the production of vascular endothelial growth factor (VEGF) by podocytes is required to maintain the adjacent glomerular endothelium in the in vivo experiments [1]. The disruption of VEGF function either through genetic or pharmaco-logic ablation results in proteinuria and renal damage with a pattern of thrombotic microangiopathy was shown in a mouse model. In the immunohistochemical staining analysis of kidney biopsy from mouse model, the complement components and im-mune complexes were negative, but fibrin staining was positive. These findings of this previous study suggests that the reduction of glomerular VEGF by bevacizumab therapy may be the pathophysiology of TMA [1]. In addition, this study demonstrated classic pathological features of (TMA) in six patients who had proteinuria after bevacizumab therapy [1]. Another previous study also reported that TMA is the main biopsy finding in bevacizumab induced proteinuria [2].

Though ICIs had been reported to induce TMA in previous studies, the incidence is very low. In addition, the patients who were diagnosed ICIs-induced TMA presented by the clinical feature of acute kidney injury (AKI) with obviously increased serum creatinine not heavy proteinuria [3,4]. The 2 patients in this report presented by asymptomatic heavy proteinuria, and did not typical symptoms of AKI such as in-creased serum creatinine level, oliguria or anuria. Base on the experiences of previous study and clinical presentation of the 2 patients, bevacizumab was discontinued priorly to nivolumab. The proteinuria recovered after discontinuation of bevacizumab. Alt-hough chemotherapy including carboplatin and paclitaxel may have nephrotoxicity, the two agents were administrated in initial 6 cycles of therapy and the proteinuria developed after cycle 20 therapy in the 2 patients. The pathological findings in both patients showed fibrin thrombi, tactoids and double-contoured appearance of the capillary wall within the glomerular capillary, and confirmed the diagnosis of TMA. Taken together, bevacizumab-induced TMA could be confirmed for the 2 patients in this report.

Previous studies reported that acute interstitial nephritis (AIN) is the most frequent pathological finding in ICI induced nephrotoxicity [2-5]. In an analysis of previous clinical study, immune-related adverse event and hypertension are independently predictive factors associated with ICIs-induced AKI [1]. Bevacizumab in combination with chemotherapy or epidermal growth factor receptor-tyrosine kinase inhibitor (EGFR-TKI) for the treatment of advanced NSCLC have been explored in several prospective clinical trials, and the incidence rate of grade 3-4 AE of proteinuria recorded in these previous trials ranged from 6 to 8% in patients who received bevacizumab therapy [6]. In two recent phase 3 trials investigating the efficacy of ICIs combined with bevacizumab, the incidence rate of grade 3-4 AE of proteinuria ranged from 2.8 to 4.8 % in the study arm with ICIs combined with bevacizumab and were not higher than that in control arm without ICIs (2.8-3.6%). Taken together, the combination of ICIs and bevacizumab did not increase risk of nephrotoxicity and proteinuria.

Point 3: Could kidney imaging, eventually US, be useful for an early diagnosis?

Response 3: In response to the comment about “Could kidney imaging, eventually US, be useful for an early diagnosis?”, we had added the kidney ultrasonography of the 2 patients to address this comment.

The kidney ultrasonography of both study patients showed no significant renal parenchymal abnormality. Therefor, it is difficult to use a kidney ultrasonography for early detection of TMA.

Point 4: The other renal functional tests are not reported.

Response 4: In response to the concern about “the other renal functional tests are not reported”, we had added a figure 3 with renal function tests data of both study patients in revised manuscript as suggested.

The changes of proteinuria, serum creatinine level and treatment agents in both patients are shown in Figure 3 in revised manuscript.

Reference

  1. Eremina, V.; Jefferson, J.A.; Kowalewska, J.; Hochster, H.; Haas, M.; Weisstuch, J.; Richardson, C.; Kopp, J.B.; Kabir, M.G.; Backx, P.H.; et al. VEGF inhibition and renal thrombotic microangiopathy. N Engl J Med. 2008 Mar 13;358(11):1129-36.
  2. Perazella, M.A.; Izzedine, H. New drug toxicities in the onco-nephrology world. Kidney Int. 2015 May;87(5):909-17.
  3. Cortazar, F.B.; Marrone, K.A.; Troxell, M.L.; Ralto, K.M.; Hoenig, M.P.; Brahmer, J.R.; Le, D.T.; Lipson, E.J.; Glezerman, I.G.; Wolchok, J.; et al. Clinicopathological features of acute kidney injury associated with immune checkpoint inhibitors. Kidney Int. 2016 Sep;90(3):638-47.
  4. Meraz-Muñoz, A.; Amir, E.; Ng, P.; Avila-Casado, C.; Ragobar, C.; Chan, C.; Kim, J.; Wald, R.; Kitchlu, A. Acute kidney injury associated with immune checkpoint inhibitor therapy: incidence, risk factors and outcomes. J Immunother Cancer. 2020 Jun;8(1):e000467.
  5. Wanchoo, R.; Karam, S.; Uppal, N.N.; Barta, V.S.; Deray, G.; Devoe, C.; Launay-Vacher, V.; Jhaveri, K.D. Cancer and Kidney International Network Workgroup on Immune Checkpoint Inhibitors. Adverse Renal Effects of Immune Checkpoint Inhib-itors: A Narrative Review. Am J Nephrol. 2017;45(2):160-169.
  6. Lee, S.H.; Lin, Y.C.; Chiu, L.C.; Ju, J.S.; Tung, P.H.; Huang, A.C.; Li, S.H.; Fang, Y.F.; Chen, C.H.; Kuo, S.C.; et al. Comparison of afatinib and erlotinib combined with bevacizumab in untreated stage IIIB/IV epidermal growth factor receptor-mutated lung adenocarcinoma patients: a multicenter clinical analysis study. Ther Adv Med Oncol. 2022 Jul 23;14:17588359221113278.

Reviewer 3 Report

Thank you very much for submitting your research article to this journal.

The cases are a report of TMA in two patients on a regimen that included an immune checkpoint inhibitor (ICI) and bevacizumab, and although proteinuria is often experienced with bevacizumab and is often followed up, the importance of performing an aggressive renal biopsy was emphasized. 

I would like to confirm the following several points.

1. regarding the statement in lines 86-88, I found that the first report of IMpower150 (Socinski, et al. NEJM 2018) has a description of adverse events of proteinuria in the supplemental material. I think it was omitted because the cited reference was a follow-up report.

2. The two cases reported were considered TMA due to bevacizumab since they improved after discontinuation. Why did you discontinue Bevacizumab? Is it a response to the protocol of the clinical trial? In fact, TMA has been reported as an immune-related adverse event in several ICIs, and there have also been reports of tubular damage caused by carboplatin and paclitaxel. 3. ICI or Bevacizumab?

3. If two cases of TMA occurred in a clinical trial, it seems relatively frequent. Both ICI and Bevacizumab have been reported as causative agents for TMA. do you think the risk is further increased by the combination?

Author Response

Response to Reviewer 3 Comments

Point 1: Regarding the statement in lines 86-88, I found that the first report of IMpower150 (Socinski, et al. NEJM 2018) has a description of adverse events of proteinuria in the supplemental material. I think it was omitted because the cited reference was a follow-up report.

Response 1: In response to this comment, we cited a right reference and re-wrote the paragraph as suggested.

In a previous trial IMPOWER150 which explored the efficacy atezolizumab combined carboplatin, paclitaxel, and bevacizumab therapy in NSCLC, the adverse events of proteinuria occurred in 12.5% of study subjects. The evaluation and management of se-rious proteinuria in the same trial were rarely reported.

Point 2: The two cases reported were considered TMA due to bevacizumab since they improved after discontinuation. Why did you discontinue Bevacizumab? Is it a response to the protocol of the clinical trial? In fact, TMA has been reported as an immune-related adverse event in several ICIs, and there have also been reports of tubular damage caused by carboplatin and paclitaxel. 3. ICI or Bevacizumab?

Response 2: In response to this comment, we discussed this point deeper and cited more references in revised manuscript.

A previous study found that the production of vascular endothelial growth factor (VEGF) by podocytes is required to maintain the adjacent glomerular endothelium in the in vivo experiments [1]. The disruption of VEGF function either through genetic or pharmaco-logic ablation results in proteinuria and renal damage with a pattern of thrombotic microangiopathy was shown in a mouse model. In the immunohistochemical staining analysis of kidney biopsy from mouse model, the complement components and im-mune complexes were negative, but fibrin staining was positive. These findings of this previous study suggests that the reduction of glomerular VEGF by bevacizumab therapy may be the pathophysiology of TMA [1]. In addition, this study demonstrated classic pathological features of (TMA) in six patients who had proteinuria after bevacizumab therapy [1]. Another previous study also reported that TMA is the main biopsy finding in bevacizumab induced proteinuria [2].

Though ICIs had been reported to induce TMA in previous studies, the incidence is very low. In addition, the patients who were diagnosed ICIs-induced TMA presented by the clinical feature of acute kidney injury (AKI) with obviously increased serum creatinine not heavy proteinuria [3,4]. The 2 patients in this report presented by asymptomatic heavy proteinuria, and did not typical symptoms of AKI such as in-creased serum creatinine level, oliguria or anuria. Base on the experiences of previous study and clinical presentation of the 2 patients, bevacizumab was discontinued priorly to nivolumab. The proteinuria recovered after discontinuation of bevacizumab. Alt-hough chemotherapy including carboplatin and paclitaxel may have nephrotoxicity, the two agents were administrated in initial 6 cycles of therapy and the proteinuria developed after cycle 20 therapy in the 2 patients. The pathological findings in both patients showed fibrin thrombi, tactoids and double-contoured appearance of the capillary wall within the glomerular capillary, and confirmed the diagnosis of TMA. Taken together, bevacizumab-induced TMA could be confirmed for the 2 patients in this report.

Previous studies reported that acute interstitial nephritis (AIN) is the most frequent pathological finding in ICI induced nephrotoxicity [2-5]. In an analysis of previous clinical study, immune-related adverse event and hypertension are independently predictive factors associated with ICIs-induced AKI [1].

Point 3: If two cases of TMA occurred in a clinical trial, it seems relatively frequent. Both ICI and Bevacizumab have been reported as causative agents for TMA. do you think the risk is further increased by the combination?

Response 3: Regarding the comment about “do you think the risk is further increased by the combination?”, we had added a paragraph to discuss it and added some references to address this comment in revised manusctipt.

Bevacizumab in combination with chemotherapy or epidermal growth factor receptor-tyrosine kinase inhibitor (EGFR-TKI) for the treatment of advanced NSCLC have been explored in several prospective clinical trials, and the incidence rate of grade 3-4 AE of proteinuria recorded in these previous trials ranged from 6 to 8% in patients who received bevacizumab therapy [6]. In two recent phase 3 trials investigating the efficacy of ICIs combined with bevacizumab, the incidence rate of grade 3-4 AE of proteinuria ranged from 2.8 to 4.8 % in the study arm with ICIs combined with bevacizumab and were not higher than that in control arm without ICIs (2.8-3.6%). Taken together, the combination of ICIs and bevacizumab did not increase risk of nephrotoxicity and proteinuria.

Reference:

  1. Eremina, V.; Jefferson, J.A.; Kowalewska, J.; Hochster, H.; Haas, M.; Weisstuch, J.; Richardson, C.; Kopp, J.B.; Kabir, M.G.; Backx, P.H.; et al. VEGF inhibition and renal thrombotic microangiopathy. N Engl J Med. 2008 Mar 13;358(11):1129-36.
  2. Perazella, M.A.; Izzedine, H. New drug toxicities in the onco-nephrology world. Kidney Int. 2015 May;87(5):909-17.
  3. Cortazar, F.B.; Marrone, K.A.; Troxell, M.L.; Ralto, K.M.; Hoenig, M.P.; Brahmer, J.R.; Le, D.T.; Lipson, E.J.; Glezerman, I.G.; Wolchok, J.; et al. Clinicopathological features of acute kidney injury associated with immune checkpoint inhibitors. Kidney Int. 2016 Sep;90(3):638-47.
  4. Meraz-Muñoz, A.; Amir, E.; Ng, P.; Avila-Casado, C.; Ragobar, C.; Chan, C.; Kim, J.; Wald, R.; Kitchlu, A. Acute kidney injury associated with immune checkpoint inhibitor therapy: incidence, risk factors and outcomes. J Immunother Cancer. 2020 Jun;8(1):e000467.
  5. Wanchoo, R.; Karam, S.; Uppal, N.N.; Barta, V.S.; Deray, G.; Devoe, C.; Launay-Vacher, V.; Jhaveri, K.D. Cancer and Kidney International Network Workgroup on Immune Checkpoint Inhibitors. Adverse Renal Effects of Immune Checkpoint Inhib-itors: A Narrative Review. Am J Nephrol. 2017;45(2):160-169.
  6. Lee, S.H.; Lin, Y.C.; Chiu, L.C.; Ju, J.S.; Tung, P.H.; Huang, A.C.; Li, S.H.; Fang, Y.F.; Chen, C.H.; Kuo, S.C.; et al. Comparison of afatinib and erlotinib combined with bevacizumab in untreated stage IIIB/IV epidermal growth factor receptor-mutated lung adenocarcinoma patients: a multicenter clinical analysis study. Ther Adv Med Oncol. 2022 Jul 23;14:17588359221113278.

Round 2

Reviewer 2 Report

This new edition  reads better.